# Enriched Fresh Pasta with a Sea Bass By-Product, a Novel Food: Fatty Acid Stability and Sensory Properties throughout Shelf Life

**DOI:** 10.3390/foods10020255

**Published:** 2021-01-26

**Authors:** Andrea Ainsa, Pedro L. Marquina, Pedro Roncalés, José Antonio Beltrán, Juan B. Calanche M.

**Affiliations:** Instituto Agroalimentario de Aragón -IA2- Universidad de Zaragoza-CITA, Miguel Servet, 177, 50013 Zaragoza, Spain; andreaainsa6@gmail.com (A.A.); pmarquin@gmail.com (P.L.M.); roncales@unizar.es (P.R.); jbeltran@unizar.es (J.A.B.)

**Keywords:** EPA/DHA, fish′s pasta, commercial shelf life, healthy pasta, Ω-3

## Abstract

A by-product from the filleting of sea bass (*Dicentrachus labrax*) was used to manufacture enriched pasta. The research aimed at achieving an increase of protein and unsaturated fatty acid contents, making innovative and healthy pasta products that are able to step up fish consumption. Two different kind of cereals were used to make fresh pasta, which were subjected to frozen storage with the addition or not of a rosemary antioxidant. The developed pastas were analyzed by physicochemical methods. Proximal analysis demonstrated an improvement of nutritional values on those of a common pasta. Fatty acid profiles, acidity indices, and TBARS (Thiobarbituric acid reactive substances) index confirmed the stability of fat and effective protection against oxidation, especially in pasta with added antioxidant. The cooking time for pasta was set at 90 s, and color parameters were modified due to the incorporation of fish in the pasta-making process. An enrichment in fatty acids ω-3 and ω-6 was also confirmed. The conversion of α-linolenic acid (ALA) in eicosapentaenoic acid (EPA) and docosahexaenoic acid (DHA) in frozen storage was detected, which remained stable during 90 days. Finally, sensory profiles of enriched pasta were found to be adequate and improved following the addition of an antioxidant due to a decrease of negative attributes associated with oxidation.

## 1. Introduction

Pasta, according to Real Decreto 2181/1975 (with last modification of 2013) [1], is a group of products obtained by the desiccation of an unfermented mass made with semolina or flour derived from durum wheat, semi-hard wheat, soft wheat, or mixtures of them and drinking water. Pasta consists mainly of carbohydrates and proteins that reach a high nutritional value [2]. Pasta can be also produced from spelt, which is a particular species of wheat (*Triticum spelta*). Spelt pasta provides a higher protein and lipid content, bran fiber, and a more desirable fatty acids profile compared to wheat [3,4]. Furthermore, all types of pasta offer great versatility, low cost, and easy preparation [5]. Therefore, pasta appears to be an excellent option to be enriched by incorporating alternative sources of nutrients, since it is a very popular food [6]. 

Then, pasta may be enriched with fish processing by-products, which implies a better use of waste generated in industrial processing, as it is used as raw material for a quality food, thus contributing to a more sustainable production system. A supplementation or replacement with fish implies an enrichment of pasta in polyunsaturated fatty acids, including those of the Ω-3 group, which would be a great opportunity to obtain the recommended daily intake of healthier fatty acids [7]. Figure 1 shows the current trends that will shape the consumption of globally appreciated pasta in the coming decades according to a recognized online platform devoted to Italian food and beverage that is specialized in trending topics studies (Italianfood) [8].

Fatty acids have important functions in the human organism and are essential nutrients for human health. Depending on the chemical structures, fatty acids are grouped in saturated (SFA), monounsaturated (MUFA), and polyunsaturated (PUFA). Those that have a higher bioactivity among all of them are both ω-3 and ω-6 PUFAs, being structural and functional components of cell membranes [9]. The α-linolenic acid (ALA; C18:3 *n*-3), eicosapentaenoic acid (EPA; C20:5 *n*-3), docosapentaenoic acid (DPA; C22:5 *n*-3), and docosahexaenoic acid (DHA; C22:6 *n*-3) are the most relevant (EFSA, 2010) [10]. Regarding health benefits for consumers, PUFAs play a fundamental role in preventing cardiovascular diseases, type II diabetes, eye diseases, arthritis, and cystic fibrosis [11]. Several studies in pasta have shown that the addition of these fatty acids affects positively the composition and nature of the lipids found in the final product, thus reducing the ratio of ω-6/ω-3 PUFA [7].

Concerning food enrichment, it should be ensured that the added nutrients reach the consumer in the best possible state. A shelf life study should be able to demonstrate that the above is achieved and confirm the nutritional contribution to a healthy diet of the resulting food. The “basic storage design” is the most used technique to monitor the shelf life of food, which consists of performing physicochemical and sensory analyses at regular intervals of time during storage [13,14]. Two main chemical reactions may compromise food stability over time when considering its shelf life as a function of the stability of its lipid components. These reactions are the hydrolysis of acyl glycerides, which is properly known as lipolysis, and those of lipid oxidation coupled with free radicals’ formation. To determine the shelf life, it is essential to know the chemical composition of the fat, the conditions under which the food is stored, and the fact that an antioxidant substance has been added [15]. 

Antioxidants are important to food preservation, as they inhibit oxidative processes. In recent years, interest in antioxidants obtained from natural sources has greatly arisen. Among them, aromatic herbs represent an excellent option due to their high antioxidant power and natural origin. Rosemary extracts (*Rosmarinus officinalis* L.) are one of the most important, and it is nowadays the only one approved by European legislation. This substance might also exert an antimicrobial effect [16,17,18] and is considered a “clean label” ingredient, i.e., clear, clean, and understandable from the point of view of food labeling [19]. This research had as purpose to enrich pasta with fish by-products in order to increase their nutritional values as well as to evaluate the stability of polyunsaturated fatty acids during frozen storage to ensure that they remain in the necessary proportions and state throughout commercial shelf life as to provide an adequate nutritional effect on consumers.

## 2. Materials and Methods

### 2.1. Raw Material

Sea bass (*Dicentrarchus labrax*) trimmings and small pieces considered as a by-product from the filleting process were provided by a local fish industry (Scanfisk^®^, Zaragoza, Spain). Regarding cereal source, semolina from durum wheat (*Triticum durum*) and spelt (*Triticum spelta*) were used. Both cereal preparations were supplied by a local company (Innova Obrador^®^, Zaragoza, Spain) dedicated to pasta making for catering and restaurants. This company has two main production lines: the first one dedicated to making simple and compound pasta from semolina of durum wheat addressed to all kind of public, and a second line to produce integral pasta using spelt semolina and bran as raw material, which is for specific food regimes or addressed to consumers with a concern for a healthier diet [20].

### 2.2. Fish Concentrates Preparation

The frozen by-product (−30 °C) from sea bass was defrosted 24 h before use. After that, skin and bones were manually removed, and quality was checked by a sensory analysis (Quality Index) developed by experts from the University of Zaragoza to ensure the suitability of raw material. The fish concentrate was produced according to the procedure described in Calanche et al. (2019) [21]. Fresh fish cuts from industrial filleting of sea bass (by-product) were selected. After that, bone and skin were removed, and the resulting flesh portions were dipped in saline solution 8% and dried in oven using slow air velocity (60 °C for 24 h). They were ground until finely uniform. In this way, two kinds of concentrate (with antioxidant or without antioxidant) were made due to the interest in the effect of antioxidant incorporation. The antioxidant used was rosemary extract powder (E-392) provided by Marbys^®^ (Barcelona, Spain), which acts as a manufacturing aid in fish concentrate.

### 2.3. Preparation of Pasta

Four types of pasta were manufactured using different formulations based on previous research studies [5,21,22,23]. Two of them were elaborated with durum wheat semolina in formulation with (Durum F + antioxidant) or without (Durum F) rosemary extract. The other two were elaborated with spelt semolina and bran to obtain pasta with a high fiber content, including rosemary extract (Spelt F + antioxidant) or not (Spelt F). The pasta formulations are shown in Table 1. They were produced with an experimental extrusion machine (Bottene, Mod. Lillodue 14057CE, Marano Vicentino, Italy) according to Calanche et al., 2019 [21]. The fresh pasta (non dessicated; F) obtained, which was named according to Spanish Regulation (Real Decreto 2181/1975) [1], was kept under frozen storage (−20 °C). For shelf life study, batches of 0.5 kg of pasta were processed for each treatment. Analyses were performed every month in each of the four types of pasta. The shelf life established was three months according to the “best before” for fresh commercial pasta.

### 2.4. Proximal Analyses

Proximal analyses (protein, fat, fiber, ash, and energy value) of pasta were determined according to AOAC, 2000; AOAC, 1990; Prosky et al., 1988; AOAC, 2005 and Merrill and Watt, 1973, respectively [24,25,26,27,28].

### 2.5. Shelf Life Study for Enriched Fresh Pasta 

#### 2.5.1. Fatty Acid Profile

A basic storage design was performed to study shelf life [10] in pasta developed throughout the determination of fatty acid profiles and chemical parameters (TBARS and acidity index). Fatty acid profiles were determined according to Bligh and Dyer, 1959 [29]. Firstly, each sample was homogenized with an Ultraturrax device (IKA-WERKE, T-25 basic) using different solvents such as chloroform, methanol, potassium chloride, and water. This mixture was centrifuged for 10 min at 4000 rpm, and the fat was extracted from the top, and afterwards, after incorporating BHT, butylated hydroxytoluene, as an antioxidant substance, solvents were evaporated with nitrogen gas. Afterwards, methylation was performed using 0.03 g of this previous fat. This fat was mixed with an intern pattern, C23:0, which does not caused interferences in the matrix. Then, 2 ml of hexane and 1 ml of potassium hydroxide saturated solution in methanol was added. The upper phase was extracted to measure. To analyze the fatty acid profile, a gas chromatograph (HP-6890 II Hewlett-Packard) was employed with a column SP-2380 (100 m × 0.25 mm × 0.20 µm). The temperature program was 140–165 °C at 3 °C/min during 10 min and 165–220 °C at 5 °C/min during 50 min. Fatty acid content was quantified as total area (%) of identified fatty acids.

#### 2.5.2. TBARS

Lipid oxidation (TBARS) was determined according to Pfalzgraf et al., 1995 [30]. A curve pattern was performed using 1, 1, 3, 3-tetrametoxipropane, TMP, to obtain the quantity of malondialdehyde (MDA). Extraction was made with 2 g of sample, and the mixture was homogenized with trichloroacetic acid (TCA) concentrate. After centrifuging the samples at 4 °C for 30 min at 4000 rpm, the reaction with thiobarbituric acid (TBA) was carried out in a thermostatic bath (J. P. Selecta, Unitronic 2000, Barcelona, Spain) at 97 °C for 20 min. After cooling the samples, absorbance was measured at 532 nm in a spectrophotometer (UNICAM, 5625 UV/VIS).

#### 2.5.3. Acidity Index

Acidity index was determined according to COPANT NTE INEN 0038, 1969 [31]. Samples were homogenized with petroleum ether, which was evaporated. Afterwards, 10 ml of neutralized ethanol was added, and samples were titrated with a solution of NaOH 0.1 N.

#### 2.5.4. Health Indexes 

The atherogenic (AI) and thrombogenic indexes (TI) were calculated using the equations of Ulbright and Soughgate (1991) [32]:(1)$$\mathrm{AI}=\frac{\left(C18:0\right)+\left(4\times C14:0\right)+\left(C16:0\right)}{\left(\mathrm{PUFA}\;n-6\;\mathrm{and}\;n-3\right)+\mathrm{MUFA}}$$
(2)$$\mathrm{TI}=\frac{\left(C14:0\right)+\left(C16:0\right)+\left(C18:0\right)}{\left(0.5\times \mathrm{MUFA}\right)+\left(0.5\times \mathrm{PUFA}\;n-6\right)+\left(3\times \mathrm{PUFA}\;n-3\right)+(\mathrm{PUFA}\;n-3/\mathrm{PUFA}\;n-6)}$$

MUFA: Monounsaturated Fatty Acids; PUFA: Polyunsaturated Fatty Acids.

### 2.6. Technological Properties

#### 2.6.1. Optimal Cooking Time Estimation

The ideal cooking time was estimated by means of an instrumental test based on a Warner–Bratzler cut test. The instrumental determination of the optimal cooking point was made by measuring the hardness of the cooked pasta using a texturometer (ANAME, TA-XT2i) with a flat Warner–Bratzler device. Hardness was described as the maximum force to the sample—expressed in kg, and the cutting effort was also determined, which was expressed in kg·s. The test was carried out by configuring the following parameters on the texturometer: pre-test speed: 2 mm/s; test speed: 2 mm/s; post-test speed: 10 mm/s; cutting distance: 15 mm; threshold strength: 10 g. After cooking, the pasta was left to reach room temperature and was always covered with damp paper to prevent drying. The time elapsed between the cooking of the pasta and the measurement of the hardness was 15 min. The hardness of each type of pasta was determined in triplicate at 30 s intervals. 

#### 2.6.2. Pasta Color

Color readings were taken from three separate points on the surface of cooked pasta (after pasta was cooked to optimal cooking time, drained, and allowed to stand for 5 min at room temperature). Color measurements were performed using a colorimeter (Minolta, CM-2002, Osaka, Japan). For the characterization of this parameter, the CIE *L***a***b** system was used, in which the color is represented by the following coordinate system: *L** (brightness), *a** (redness), and *b** (yelowness). For each sample, readings were taken six times. The color variation produced by the addition of different ingredients was calculated with the total color difference (ΔE) between pasta standard with fish adedd (Durum F and Spelt F) and pasta with antioxidant added (Durum F and Spelt F with antioxidants), which is determined by the following equation:(3)$$\Delta E=\;\sqrt{{\left(\Delta L^{*}\right)}^{2}+{\left(\Delta a^{*}\right)}^{2}+{\left(\Delta b^{*}\right)}^{2}}$$

Δ*L* = *L** Fish pasta−*L** Standard pasta; Δ*a* = *a* Fish pasta−*a** Standard pasta and Δ*b* = *b** Fish pasta −*b** Standard pasta.

#### 2.6.3. Cooking Losses

The standard method (AACC 66-50) was carried out to estimate this technological parameter. In brief, 3 g of each kind of pasta developed were immersed in 180 mL of water and boiled during their respective optimal cooking times that previously had been estimated [33]. The water resulting after heating was collected in crucibles and allowed to evaporate in a stove at 105 °C until reaching a constant weight (24 h). The dry residue was weighed on an analytical balance and determined as a percentage of the total weight of the pasta before cooking. This analysis was developed in triplicate by each type of developed pasta.

### 2.7. Sensory Analysis

A sensory trained panel with previous experience on the assessment of quality of pastas was used. A quantitative descriptive analysis (QDA) was used employing a non-structured scale of 10 cm to evaluate the intensity of each sensory attribute stablished for pasta with fish according to previous research where a group of 16 sensory attributes was produced by sensory profiling technique (typical aroma, flour odor, fish odor, characteristic color—yellow or brown, homogeneity aspect, elasticity, hardness, disintegrability, granularity, pastiness, chewiness, characteristic flavor, flour flavor, fish flavor, rancidity flavor, and after taste) [21]. Two sessions that complied with the provisions of ISO 8589:2007 [34] were held to establish sensory profiles in developed pastas both in the beginning and at the end of the experiment, according to criteria of sensory assessors following the recommendations of Standard UNE-EN ISO 13299:2017 [35], in order to detect and differentiate adequately changes occurred over storage time. In these sessions, pastas were prepared by cooking in water boiling (100 °C) during times established in previous studies. Subsequently, the tasters were served without any type of accompaniment at a temperature of 60 °C, at all times complying with the recommendations of Standard UNE-ISO 6658:2019 [36].

### 2.8. Statistical Analyses

Results were analyzed with descriptive and inferential statistics using a XLSTAT Version 2016 (Addinsoft^©^, Paris, France). First, a univariate analysis was carried out to verify the normality of the data and detect outliers’ values. Once the previous step was executed, statistical analysis was done by two-way ANOVA (treatment/storage time/interaction) and using the multiple comparison test a posteriori Fisher with a 95% confidence interval to establish differences among means. Relationships between variables were studied applying a parametric correlation analysis (Pearson). Finally, Principal Component Analysis (PCA) was performed, as multivariate analysis methods based on covariance matrix (n) from data in order to achieve a comprehensive understanding of the study and establish relationships among the variables and treatments considered. In the sensory study, panel analyses were done to establish the performance and reliability of the results, checking the panel’s performance as well as its discriminative power. For frozen storage, ANOVA was done to obtain those parameters with significant differences over time and among developed pasta. Finally, characterization of the product was carried out from the square cosine method to get sensorial profiles for each type of pasta, which was presented in a plot (PCA) where trust ellipses (95%) were drawn to compare among pasta and sampling times.

## 3. Results 

### 3.1. Characterization of the Developed Pastas

Table 2 shows proximal analyses for both types of pasta. Durum pasta had 2.89% fat on average while spelt pasta reached 4.97%. Spelt pasta had a fiber content almost nine times higher than that of durum wheat. Significant differences (*p* < 0.05) were also found in ash percentages. The energy of durum pasta (≈353 kcal/100 g) was significantly (*p* < 0.05) higher than that of spelt pasta (≈330 kcal/100 g).

### 3.2. Fatty Acid Profile throughout Storage Time

Relevant fatty acids were selected for quantification in the different developed pastas. They were classified in two groups; those provided mainly by fish (C16:0, palmitic; C18:3 n-3, ALA; C20:5 n-3, EPA; and C22:6 n-3, DHA) and those proceeding mainly from cereals (C18:1 n-9, oleic; and C18:2 n-6, linoleic) [37,38]. 

Fatty acid profiles along the storage time are shown in Table 3. Durum pasta (Durum F) showed a significant decrease (*p* < 0.05) in ALA and DHA amounts, which was probably due to lipid oxidation, while EPA’s quantities remained stable. Significant changes (*p* < 0.05) were detected in fatty acid composition when rosemary extract was added to durum pasta (Durum F + antioxidant). Oleic acid increased over time, while ALA revealed a decrease, especially at 90 days. However, EPA and DHA values showed an increase (*p* < 0.05) throughout refrigerated storage. Only DHA demonstrated significant differences (*p* < 0.05) between treatments (with and without antioxidant) from day 60. 

Spelt pasta showed a significant decrease (*p* < 0.05) in ALA quantity, too, as well as an increase in EPA and DHA amounts. In general, spelt pasta demonstrated a higher stability of unsaturated fatty acids over storage than durum pasta. This behavior might be related to the different flours of each formulation. Likewise, the addition of rosemary extract on spelt pasta (Spelt F + antioxidant) had a positive effect against oxidation in EPA and DHA; they maintained higher concentrations (*p* < 0.05) from the beginning of the experiment and throughout their commercial shelf life (best before). 

### 3.3. Ratios and Chemical Indices throughout Storage Time

The main ratios and chemical indices for the different developed pastas (durum and spelt) with or without rosemary extract are summarized in Table 4 and Table 5. The percentage of MUFA in spelt pasta showed a significant increase (*p* < 0.05) throughout storage time, especially in Spelt F + antioxidant—that should be related to the release of oleic acid, which was confirmed by the acidity index. This index for spelt pasta remained stable for most of the sampling times, showing only a significant (*p* < 0.05) increase at the end of the study.

It appeared evident that P/S (PUFA/SFA) ratios remained stable (around ≈2) in all pastas. The ω6/ω3 ratios showed an increase in “Spelt F”. On the other hand, mg ω3/100 g decreased in most of the treatments over time due to fatty acid oxidation. In fact, adding a rosemary extract to spelt pasta (Spelt F + antioxidant) caused an increase from 391 to 459 at the final sampling time. This behavior could be the result of the release of fatty acids over time after a period of stability due to the antioxidant effect of rosemary, which was confirmed by the increase of the acidity index at 90 days. Durum pasta showed ratios between 178 and 304 mg/100 g, while spelt pasta had even higher values (251–459 mg/100 g). 

Durum F + antioxidant pasta exhibited TBARS values (in average, 1.09) that were significantly lower than Durum F (1.31) throughout the study, thus demonstrating the effectivity of rosemary extract to prevent lipid oxidation. A significant increase (*p* < 0.05) from day 30 to day 60 was found in Durum F. However, from day 60 to day 90, this behavior changed, and a significant decrease (*p* < 0.05) was observed. In Durum F + antioxidant pasta, an increase between 30 and 60 days was seen, too. Regarding Spelt F pasta, its TBARS values increased at day 30. Spelt + antioxidant pasta values remained stable from day 0 to day 60. Nevertheless, it showed an increase (*p* < 0.05) at day 90. A synergy between rosemary extract and phenolic compounds with antioxidant activity present in the spelt bran could have reduced TBARS values in Spelt + antioxidant pasta. 

Values obtained for AI were around 0.3, while they were between 0.4 and 0.5 for TI. Both parameters showed a significant increase in Durum F at day 90. The TI value decreased significantly at day 90 in Spelt pasta + antioxidant.

### 3.4. Principal Components Analysis (PCA) 

In order to reach an overview of the behavior of the main fatty acids and their relationship with chemical parameters, a PCA was carried out as a comprehensive study for a better interpretation of the results. The PCA plot for durum pasta including the quality parameters assessed (fatty acids, TBARS, and acidity index (%)) throughout storage time (90 days) is shown in Figure 2A. The two first principal components (F1 and F2) accounted for 70.28% of total variability. The first component separated ALA from the rest of the fatty acids and confirmed the observed conversion of this fatty acid to EPA and DHA, which were both associated with the final time of storage (day 90), especially in the Durum F + antioxidant pasta. The TBARS index was well correlated with oleic and linoleic acids and located close to day 60 and day 90. This could indicate that these fatty acids have been the most vulnerable to oxidation during refrigerated storage, since they were found in considerable amounts in developed pasta (≈20–40 %). 

Regarding spelt pasta, the PCA plot, which explained 76.75% of the total variability, is shown in Figure 2B. As in the previous case of durum pasta, F1 separated ALA from the rest of fatty acids, which confirmed once again the conversion of ALA to EPA and DHA.

The evolution of the shelf life over frozen storage can be also appreciated, which shifted from the left to the right side of the plot. Four distinct groups were formed, each one corresponding to a different sampling time. The acidity index was located close to linoleic acid, thus indicating that it could be liable to oxidation, even though the TBARS index for this sampling time (Day 90) did not show any significant difference. 

Concerning the antioxidant activity on fatty acids, Spelt F + antioxidant pasta demonstrated an effective protection for the whole 90 days established as commercial shelf life for this study. Hence, the rosemary extract addition seemed to be an effective treatment to avoid the loss of unsaturated fatty acids due to oxidation and therefore to warrant enough quantities of EPA and DHA in the enriched pasta.

### 3.5. Optimal Cooking Time Estimation

As shown in Figure 3, hardness decreased as the cooking time increased until it reached an inflection point. The optimal cooking time was considered to be the value that showed such behavior. Based on the above, the optimal cooking time for the durum and spelt pasta was 90 s, since significant decreases were found between 90 and 120 s in all assayed pastas. According to the company (Innova Obrador SL) instructions for use, an optimal cooking time for fresh basic pasta (durum and spelt) is established at 120 s.

### 3.6. Color Pasta

The two types of pasta were analyzed separately, since they presented different colors. Figure 4A shows the difference between pasta with and without antioxidant in each parameter. The luminosity increased significantly with the addition of the antioxidant, while the red (*a**) and yellow (*b**) indices decreased significantly. Spelt pasta is shown in Figure 4B. In this case, the only parameter that demonstrated a significant increase when incorporating the antioxidant was the yellow index (*b**). The overall index (ΔE) showed values for durum and spelt pasta of 3.79 and 1.94, respectively.

### 3.7. Cooking Losses

Cooking losses according to the optimal times previously estimated are shown in Figure 5. Durum pasta had a greater cooking loss than spelt pasta. Furthermore, the cooking loss in Spelt F with antioxidant was significantly higher than in the Spelt F without antioxidant.

### 3.8. Sensory Profiles

Sensory profiles based on the square cosines analysis of the different pastas using QDA at initial (day 0) and final sampling time (day 90) are shown in Figure 6. An ANOVA (*p <* 0.05) demonstrated that only eight of a total of 16 attributes initially selected were used to draw plots on the basis of their discriminatory capacity. On Day 0, the different pasta types were well discriminated among them relating to their composition (durum and spelt), collecting 72.34% of the total variation in F1 (Figure 6A). However, some similarities between pasta with or without the addition of rosemary extract were detected; in fact, both kinds of durum pasta demonstrated similar profiles, in which elasticity (*p <* 0.01) was highlighted as the common key attribute, differentiating each other by the pastiness degree (*p <* 0.05). Spelt F (SF) pasta was characterized by its chewiness (*p* < 0.05) and granularity (*p* < 0.05), whereas Spelt F antioxidant (SA) showed a characteristic intense color (7/10) with flour flavor and odor, besides a moderate after-taste (5/10).

All pastas were discriminated from each other in both components F1 and F2 with 88.28% of the variance explanation (Figure 6B). Fish odor and elasticity predominated in Durum F (DF), while a homogeneity resembling common pasta was perceived in Durum F + antioxidant (DA). Meanwhile, Spelt F (SF) was characterized by fish flavor and after-taste, and Spelt F + antioxidant” (SA) stood out by its granularity in mouth and a typical color, such as that of common spelt pasta. 

Addition of the antioxidant in pasta formulation improved the sensory quality, since unpleasant oxidative odors and flavor related to fish were reduced. However, it should be noted that TBARS values on the final day of the study (day 90) were low. In this study, all pastas showed low values in negative sensory attributes such as rancidity, off flavor, and pastiness.

## 4. Discussion 

### 4.1. Characterization of the Developed Pastas

In comparison with a common pasta, the addition of fish concentrate resulted in significant rises (*p* ≤ 0.05) of the average fat content in both pasta types (2.73% durum and 4.97 % spelt; Table 2) with respect to a mean value of 1.44% for common pasta [38]. 

Protein percentage reached 19.55% on average [5], being significantly higher than the medium amount for common wheat pasta (12.50%) [7,22]. In addition, an important decrease was found in carbohydrates with respect to common pasta, reaching values of only 62.43% in Durum F and 51.59% in Spelt F pastas. Fiber composition was increased solely in spelt pasta due to the presence of bran in its formulation. Relating the ash content, durum had the highest amounts, which may be due to the incorporation of salt (NaCl) in their formulations. Energy values for durum pasta with added fish were higher than those reported by Devi (2013) and Desai et al. (2018) in similar foods [3,21].

### 4.2. Fatty Acid Profile and Chemical Parameters

The “best before” consumption label for fresh pasta kept at low temperature (< 4 °C) was established at 90 days according to the information supplied by industry companies [39]. In this study, a conversion from ALA to EPA and DHA happened over time along the fresh pasta shelf life, thus improving the availability of these fatty acids to offer health benefits [39]. Our findings were in agreement with results from a previous study on the stability of fatty acids in extruded food developed by Glodde et al. (2018) [40]. 

The ideal ratio for ω6/ω3 PUFA in an appropriate healthy diet is considered to be around 4:1 [41]. In contrast, our study found values higher than 4.0 in some cases at day 90 that exceeded this recommendation. However, these results agreed with ratios established by previous studies for pasta with added fish [7,21,23]. With respect to the ratios mg ω3/100 g, the EFSA established a recommendation of 300 mg [42] for the ω3 mg/100 g ratio, which is higher than the value of 90 mg/100 g suggested by the NHI - National Institute of Health- (2015) [43]. Higher values than those recommendations were found in this study. Consequently, our findings demonstrated that enriched pasta with fish might represent an adequate source to get the recommended daily intake of PUFA (250 mg/day) and so, to achieve a healthier diet [10].

Regarding health lipids indices (AI and TI), results were in agreement with previous reports for fish [44,45]. Therefore, values obtained could be considered as low with respect to other foods such as milk, with values around 1 for both indexes [46]. Diets with a low AI and TI values might reduce the potential risk of coronary heart disease (CHD) [32,46,47].

Lipolysis are oxidative reactions catalyzed by lipases under certain conditions, which are favored by divalent cations found in the pericarp of the bran [48,49]. One of the main reasons for the loss of quality of pasta with fish is lipid oxidation, which results in a fatty acid decrease and the generation of unwanted secondary compounds such as malonic aldehyde [40,50]. According to the above, TBARS results were in agreement with results obtained by Glodde et al. (2018) [40], who studied ω-3 fatty acids stability in other fatty foods. The increase of TBARS in spelt pasta with antioxidants at day 90 could be explained by the different milling fractions of cereal, since bran has the highest phenolic content while the endosperm possesses the lowest amount; besides that, iron (II) chelation ability is greater in the first. So, free radicals and reactive oxygen species (ROS) lower their oxidation power when bran fraction increases [51]. TBARS did not exceed in any case the acceptable limit of 1.5 mg/kg MDA in fish or meat to generate the sensory perception of oxidation [52].

Our findings on antioxidant incorporation confirmed the results of previous researchers that have used various species of fish to produce enriched pasta with sea bass concentrate, showing proven health benefits [23,40].

### 4.3. Technological Properties

The optimal cooking time showed the decrease in hardness and cooking time when adding fish. For durum pasta, the yellow index (*b**) is one of the most influential parameters for the acceptability of the pasta and therefore, the results are beneficial from a technological point of view [53]. According to previous studies, the luminosity of this type of pasta decreased with the incorporation of fish compared to a basic durum pasta [5]. In spelt pasta, the addition of flour and spelt bran in its composition gives it a darker color that can mask to some extent the effect produced by fish concentrate on color. The effect of the presence of antioxidant was higher in durum pasta than in spelt pasta. Overall indices (ΔE) of pasta with fish and antioxidant added increased significantly (*p* < 0.05) as indicative of the influence of the rosemary extract over lightness in the color study. However, ΔE values obtained in both types of pasta (3.79 and 1.94) were not more than 5.0, and these changes cannot probably be seen with the naked eye [5]. With respect to the cooking losses, it is considered that the cooking losses of optimal quality pasta should not exceed 8%. Therefore, the developed pastas present optimal values of cooking loss [54]. Furthermore, durum pasta presented higher values than spelt, which was possibly due to the higher fiber content of the spelt formulation [55,56].

### 4.4. Sensory Analysis 

The main differences between durum and spelt-enriched pastas were related to color. A previous study developed by Calanche et al. (2019) found that the addition of sea bass concentrate (10%) to pasta from durum wheat caused some sensory changes—mainly in fishy odor and a decrease in the intensity of typical yellow color [21]. In this sense, the sensory profile of durum pasta was in agreement with that obtained by Devi (2013), who developed semolina pasta with partial replacement with fish in their formulations [22]. With respect to spelt pasta, the attributes found were similar to those obtained for common spelt pasta (brown), as reported by Marconi et al. (2002). These authors indicated that spelt pasta was not as yellow (lower b*) as durum wheat pasta, which was related to the lower yellow pigmentation of spelt flour. However, the relative lack of yellow color coupled with a redder tinge of bran in spelt pasta was not a problem for consumers, since users of wholemeal products or organic foods are prepared to accept pasta that is not amber yellow [20]. In addition to color, both the granularity and after-taste showed differences among pastas, which was most probably due to the presence of bran in spelt pasta.

All pastas showed a low valuation in negative attributes such as rancidity flavors and undesirable odors. TBARS values were lower than 2 mg/kg MDA [52], which represents the limit for an acceptable sensory perception before rejection. This explained the low values in rancidity flavor. These findings agreed with results obtained in previous studies [20,22], which reported that the perception of an uncommon flavor was described in pasta enriched with ω3 fatty acids, although it did not refer to fish. As can be seen in Figure 3, both fish odor and fish flavor increased over time, having an important weight for sensory profiles of enriched pasta at the final time (90 days) of the commercial shelf life. Overall, all the developed pastas had adequate sensory profiles, which remained unaltered throughout their commercial shelf life.

## 5. Conclusions

Pasta enriched with a sea bass by-product represents an excellent source of protein, fiber in the case of spelt pasta, due to the presence of bran, and, above all, polyunsaturated fatty acids of the Ω-3 type. The pastas developed with fish concentrate showed a stability of fat and effective protection against the oxidation of unsaturated fatty acids during the commercial shelf life studied (this study brought to light the conversion of ALA in EPA and DHA in 90 days), especially in those with the addition of a rosemary extract. Unsaturated fatty acids, especially EPA and DHA, remained in satisfactory quantities during the commercial shelf life of the developed pastas and furthermore during the course of frozen storage. Finally, the sensory profiles of enriched pastas with sea bass were most adequate and improved with the addition of a rosemary extract in relation to a decrease of attributes regarded as negative and associated with rancidity.

## Figures and Tables

**Figure 1 foods-10-00255-f001:**
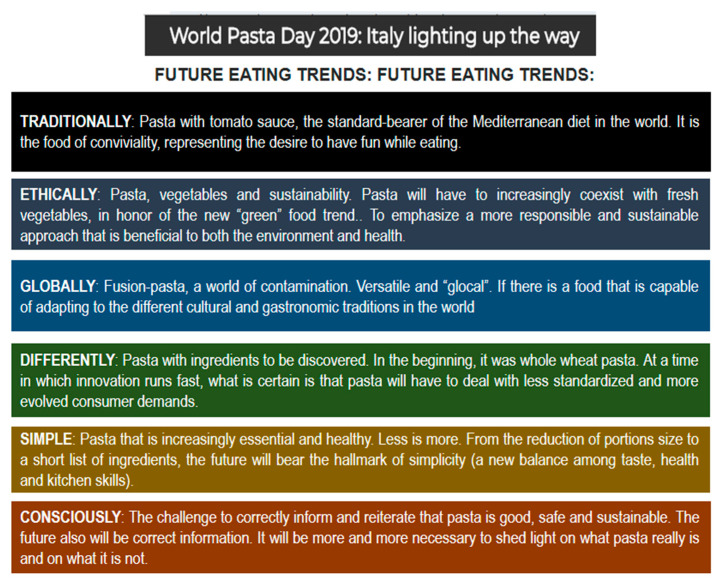
Trends that will shape the consumption of this globally appreciated food in the coming decades [12].

**Figure 2 foods-10-00255-f002:**
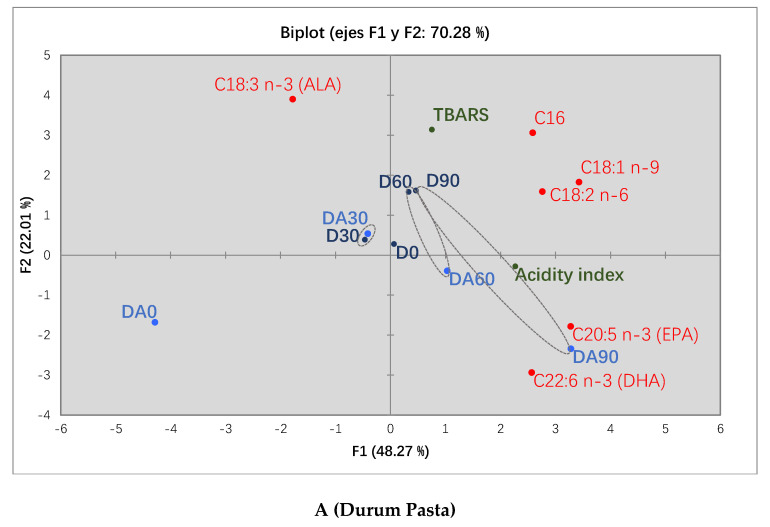
Principal Component: Analysis of developed pasta with and without antioxidant added over storage time. D0: Durum F day 0, D30: Durum F day 30, D60: Durum F day 60; D90: Durum F day 90, DA0: Durum F + antioxidant day 0, DA30: Durum F + antioxidant day 30, DA60: Durum F + antioxidant day 60, DA90: Durum F + antioxidant day 90. S0: Spelt F day 0, S30: Spelt F day 30, S60: Spelt F day 60; S90: Spelt F day 90, SA0: Spelt F + antioxidant day 0, SA30: Spelt F + antioxidant day 30, SA60: Spelt F + antioxidant day 60, SA90: Spelt F + antioxidant day 90. (**A**) Durum Pasta; (**B**) Spelt pasta.

**Figure 3 foods-10-00255-f003:**
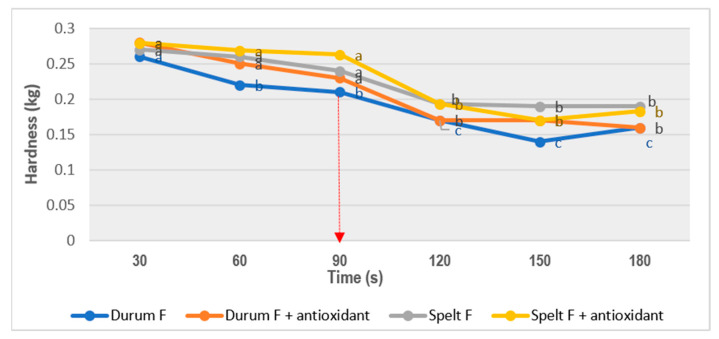
Optimal cooking time for both types of pasta. Distinct letters indicate significant difference (*p* < 0.05) among the cooking times for each kind of pasta.

**Figure 4 foods-10-00255-f004:**
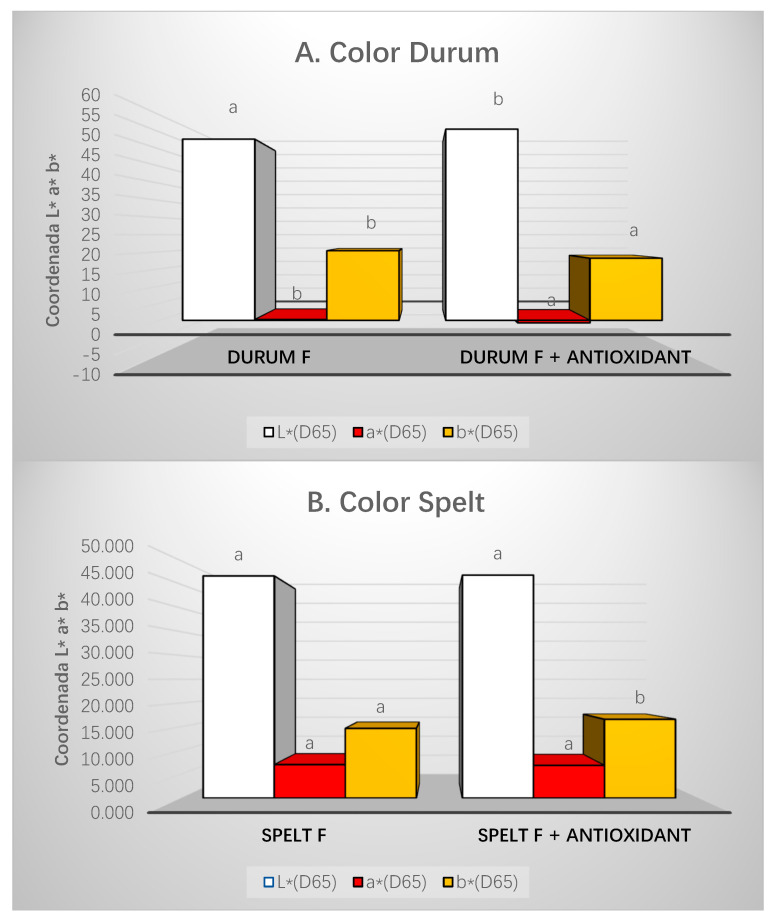
Color parameters in different types of pasta. Distinct letters indicate significant difference (*p* < 0.05) among the different parameters between treatments. (**A**) Color Durum; (**B**) Color Spelt.

**Figure 5 foods-10-00255-f005:**
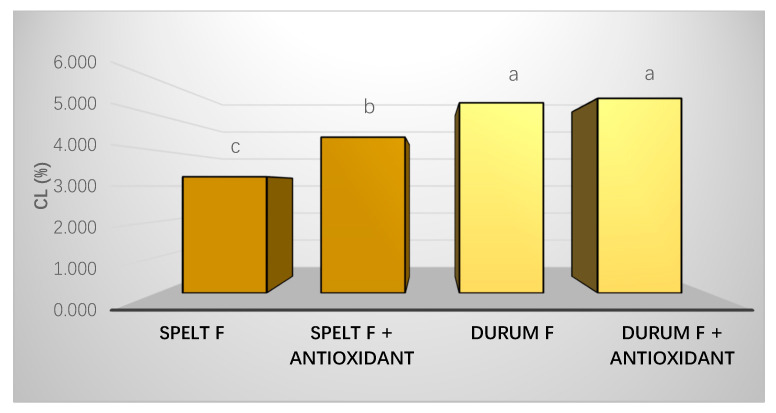
Cooking losses (%) for both types of pasta. Distinct letters indicate significant difference (*p* < 0.05) among each formulation. Two types of pasta are shown in different colors.

**Figure 6 foods-10-00255-f006:**
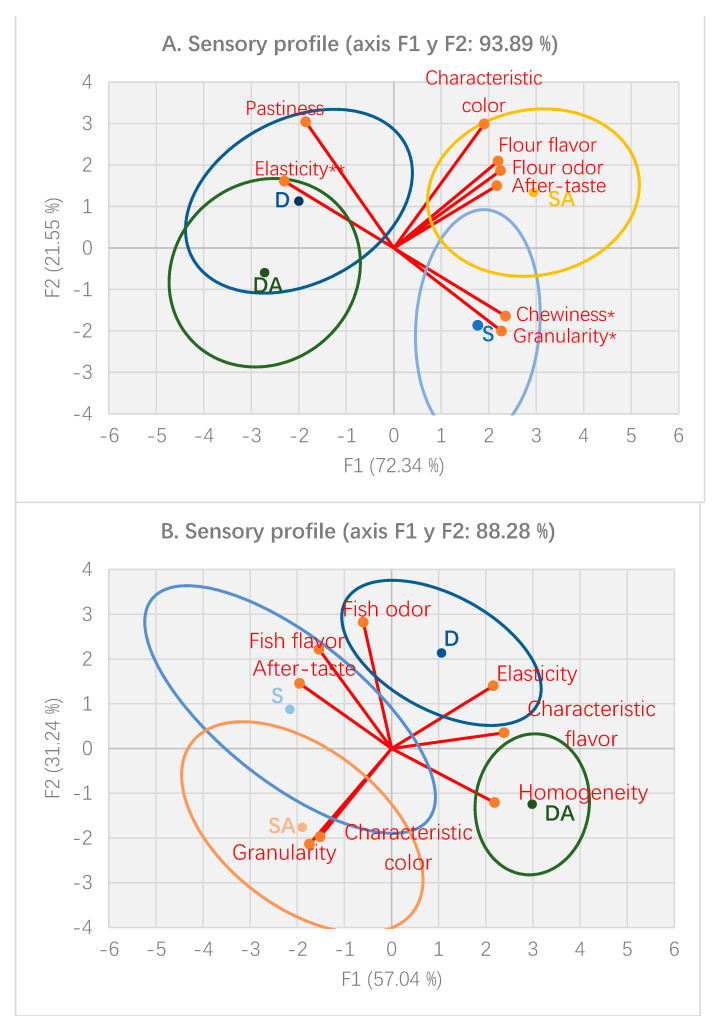
(**A**) Sensory profile at day 0 and (**B**) Sensory profile at day 90 from square cosines provided by sensory trained assessors valuations. D: Durum F, DA: Durum F + antioxidant, S: Spelt F, SA: Spelt F + antioxidant. * Attribute showed significant differences *(p* < 0.05). ** Atribute showed significant differences (*p <* 0.01)

**Table 1 foods-10-00255-t001:** Composition of formulations.

Durum F	%	Durum F with Antioxidant	%	Spelt F	%	Spelt F with Antioxidant	%
Durum wheat	64.5	Durum wheat	64.5	Spelt wheat	55	Spelt wheat	55
Fish concentrate	10	Fish concentrate and antioxidant	10	Spelt bran	10	Spelt bran	10
Aromatic herbs and spices	0.5	Aromatic herbs and spices	0.5	Fish concentrate	10	Fish concentrate and antioxidant	10
Water	25	Water	25	Water	25	Water	25

**Table 2 foods-10-00255-t002:** Proximal analysis of pastas from different cereals with fish concentrate added.

	*Triticum durum*	*Triticum spelta*
	**Durum F**	**Spelt F**
Moisture	9.50 ± 1.10	11.50 ± 1.01
Fat	2.89 ± 0.28 a	4.80 ± 0.27 b
Protein	19.60 ± 0.10	19.65 ± 0.09
Fiber	1.27 ± 0.21 a	9.05 ± 0.35 b
Ash	4.60 ± 0.10 c	3.21 ± 0.43 a
Carbohydrates	62.43 ± 0.23 b	51.59 ± 0.09 a
Energy values (Kcal/100 g)	352.95 ± 0.32 b	328.15 ± 3.35 a

Average values (*n* = 3) and standard deviation (SD) obtained from triplicated measures. Different letters in the same rows indicate significant differences (*p* ≤ 0.05) among treatments.

**Table 3 foods-10-00255-t003:** Fatty acids profiles of durum and spelt pastas with or without antioxidants.

		Storage Time			
**Treatments**		**0**	**30**	**60**	**90**
	**Durum F**				
	C16:0	15.76 a B	15.63 a B	15.75 a B	16.99 b B
	C18:1 *n*-9	25.24 ab C	25.04 a B	25.38 ab C	25.71 b D
	C18:2 *n*-6	34.09 ab C	34.10 ab C	34.40 b C	32.68 a B
	C18:3 *n*-3 (ALA)	3.91 b B	3.89 b B	3.90 b B	3.71 a B
	C20:5 *n*-3 (EPA)	2.25 B	2.23 B	2.27 B	2.23 B
	C22:6 *n*-3 (DHA)	3.60 b D	3.46 b C	3.51 b C	3.13 a A
	**Durum F + Antioxidant**				
	*C16:0*	13.95 a A	16.04 b B	15.75 b B	15.93 b C
	C18:1 *n*-9	22.11 a A	24.93 b B	25.29 c C	26.06 d E
	C18:2 *n*-6	29.55 a A	35.65 c D	34.39 b C	34.61 b C
	C18:3 *n*-3 (ALA)	3.46 ab B	3.92 b B	4.03 b B	1.40 a A
	C20:5 *n*-3 (EPA)	2.08 a A	2.24 b B	2.47 c C	2.53 c D
	C22:6 *n*-3 (DHA)	3.25 a B	3.47 b C	4.16 c E	4.23 c E
		**0**	**30**	**60**	**90**
	**Spelt F**				
	C16:0	14.02 A	14.75 AB	15.19 AB	15.84 B
	C18:1 *n*-9	26.92 a A	28.91 b B	28.74 b B	29.17 b B
	C18:2 *n*-6	31.06 a A	33.90 b C	34.29 b C	35.42 c D
	C18:3 *n*-3 (ALA)	3.58 b B	3.80 b B	3.83 b B	1.56 a A
	C20:5 *n*-3 (EPA)	1.92 A	1.96 A	1.87 A	1.87 A
	C22:6 *n*-3 (DHA)	2.45 A	2.61 B	2.43 A	2.44 A
	**Spelt F + Antioxidant**				
	C16:0	13.99 a A	15.25 b AB	15.40 b AB	14.90 b AB
	C18:1 *n*-9	26.57 a A	28.35 b B	28.85 c B	33.29 d C
	C18:2 *n*-6	30.29 a A	32.46 b B	32.41 b B	37.72 c E
	C18:3 *n*-3 (ALA)	3.42 b B	3.69 b B	1.91 ab A	1.15 a A
	C20:5 *n*-3 (EPA)	2.03 a B	2.05 ab B	2.13 b B	2.48 c C
	C22:6 *n*-3 (DHA)	2.84 a B	2.64 a B	2.74 a B	3.66 b C

Lower case letters within each row for the same pasta with or without antioxidant show significant differences (*p* < 0.05) in sampling times inside each treatment. Capital letters within columns show significant differences (*p* < 0.05) between treatments (with or without antioxidant) throughout the shelf life for each group of pasta (durum and spelt).

**Table 4 foods-10-00255-t004:** Chemical indices and ratios for durum pasta.

		Storage Time			
**Treatments**		**0**	**30**	**60**	**90**
	**Durum F**				
	%SFA	20.80 a A	22.82 c B	21.66 b B	22.17 bc B
	%MUFA	31.91 B	31.61 B	31.97 B	32.76 B
	%PUFA	46.32 b C	45.58 b B	46.28 b C	44.14 a B
	P/S ratio	2.23 c B	2.00 a A	2.14 b B	1.99 a A
	ω6/ω3 ratio	3.47 A	3.51 A	3.39 B	3.45 A
	mg ω3/100 g	241.20 c B	240.01 c B	212.27 b B	178.55 a A
	TBARS	1.13 a A	1.14 a A	1.70 c C	1.28 b B
	Acidity Ind.	0.16	0.15	0.16	0.17
	AI	0.31 a A	0.31 a A	0.31 a A	0.34 b B
	TI	0.49 a A	0.49 a A	0.49 a A	0.54 b B
	**Durum F + antioxidant**				
	%SFA	23.19 c B	21.22 b A	20.71 a A	20.96 a A
	%MUFA	28.07 a A	31.47 b B	32.06 b B	35.33 c C
	%PUFA	40.41 a A	47.31 c D	47.18 c D	43.71 b B
	P/S ratio	1.74 a A	2.23 b B	2.28 b B	2.09 b A
	ω6/ω3 ratio	3.29 a A	3.58 ab A	3.13 a A	4.71 b B
	mg ω3/100 g	289.52 c B	304.90 c C	243.10 b B	196.42 a A
	TBARS	1.06 a A	1.01 a A	1.21 b B	1.10 ab A
	Acidity Ind.	0.14 a	0.13 a	0.14 a	0.18 b
	AI	0.32 A	0.31 A	0.31 A	0.31 A
	TI	0.50 A	0.49 A	0.49 A	0.48 A

SFA: Saturated Fatty Acids. MUFA: Monounsaturated Fatty Acids. PUFA: Polyunsaturated Fatty Acids. UFA: Unsaturated Fatty Acids. P/S ratio: PUFA/SFA ratio. AI: Atherogenic index. TI: Thrombogenic index. Lowercase letters within rows show significant differences in sampling times inside each type of pasta (*p* < 0.05). Capital letters within columns show significant differences (*p <* 0.05) between treatments (with or without antioxidant) throughout shelf life for each group of pasta (durum and spelt).

**Table 5 foods-10-00255-t005:** Chemical indices and ratios for spelt pasta.

		Storage Time			
**Treatments**		**0**	**30**	**60**	**90**
	**Spelt F**				
	%SFA	19.34 A	19.66 A	20.29 B	20.45 B
	%MUFA	33.58 a A	35.97 b B	35.59 b B	37.52 c C
	%PUFA	40.84 a A	44.37 b B	44.09 b B	42.53 ab B
	P/S ratio	2.11 A	2.26 B	2.17 A	2.08 A
	ω6/ω3 ratio	3.74 a A	3.87 a A	4.13 b B	6.49 c C
	mg ω3/100 g	335.96 b B	377.46 c B	348.00 b B	251.82 a A
	TBARS	1.33 a B	1.41 b B	1.40 b B	1.24 a A
	Acidity Ind.	0.12 a	0.10 a	0.12 a	0.21 b
	AI	0.29	0.29	0.30	0.31
	TI	0.45 A	0.44 A	0.45 A	0.44 A
	**Spelt F + Antioxidant**				
	%SFA	18.66 a A	21.55 c	20.59 b B	18.67 a A
	%MUFA	33.22 a A	35.50 b B	36.10 c B	39.22 d C
	%PUFA	41.00 A	42.98 A	41.51 A	42.11 A
	P/S ratio	2.20 b A	1.99 a A	2.02 a A	2.26 b B
	ω6/ω3 ratio	3.56 a A	3.73 ab A	4.50 ab B	4.82 b B
	mg ω3/100 g	391.34 a B	377.41 a B	392.57 a B	459.51 b C
	TBARS	0.95 a A	1.05 a A	0.99 a A	1.23 b A
	Acidity Ind.	0.13 a	0.11 a	0.13 a	0.19 b
	AI	0.30 b	0.31 b	0.31 b	0.29 a
	TI	0.46 b A	0.47 b B	0.46 b A	0.42 a A

SFA: Saturated Fatty Acids. MUFA: Monounsaturated Fatty Acids. PUFA: Polyunsaturated Fatty Acids. UFA: Unsaturated Fatty Acids. P/S ratio: PUFA/SFA ratio. AI: Atherogenic index. TI: Thrombogenic index. Lowercase letters within rows show significant differences in sampling times inside each type of pasta (*p* < 0.05). Capital letters within columns show significant differences (*p* < 0.05) between treatments (with or without antioxidant) throughout shelf life for each group of pasta (durum and spelt).

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
