# Peer review of "Enriched Fresh Pasta with a Sea Bass By-Product, a Novel Food: Fatty Acid Stability and Sensory Properties throughout Shelf Life"

_foods, 2021, doi:10.3390/foods10020255_

Round 1

Reviewer 1 Report

The paper titled “ENRICHED FRESH PASTA WITH A SEA BASS BY-PRODUCT, A NOVEL FOOD: QUALITY, STABILITY AND SENSORY PROPERTIES THROUGHOUT SHELF LIFE” aimed to formulate a novel pasta enriched with a sea bass by-products.

Authors performed on novel pasta samples proximate analyses, a shelf-life study and a sensory analysis test. However, in the title, Authors wrote the term “quality”, but analysis concerning main pasta quality characteristics are completing missing. I retain that Authors must include and discuss the following pasta characteristics before a possible re-submission of the paper:

-color of pasta (prior and after cooking)

-cooking behavior including optimal cooking time, cooking loss, water absorption capacity

-texture properties of pasta (before and/or after cooking)

-properties of the protein network following sea bass by-product inclusion

-maybe also the chemical composition prior and after cooking should be of interest

Author Response

Question

Response

Page

Referee 1

1.1 Authors performed on novel pasta samples proximate analyses, a shelf-life study and a sensory analysis test. However, in the title, Authors wrote the term “quality”, but analysis concerning main pasta quality characteristics are completing missing.

Thank you very much for your recommendation. The title of work has been change, really the quality term is a general term. However, we focused on fatty stability, especially, about fatty acids. In fact, complementary analyses has been included in the manuscript to provide more specific information about technological properties.

P.1/p.5/6/14/15/17

1.2 Color of pasta (prior and after cooking)

Thank you very much for your advice. Although our aim is studied the fatty stability, color analysis has been added as quality parameter.

p.5/6/14/15/17

1.3 Cooking behavior including optimal cooking time, cooking loss, water absorption capacity

Thank you very much for your reccomendation. We have added optimal cooking time and cooking lost as quality parameter.

p.5/6/14/15/17

1.4 Texture properties of pasta (before and/or after cooking)

Thank you very much for your advice. Although we have not added a complete texture profile because the aim of this work is different, we have included the warner test to obtain the optimal cooking time.

p.5/6/14/15/17

1.5 Properties of the protein network following sea bass by-product inclusion

Thank you for your reccomendation. However, the scope of this work is far from this topic, due to limitations of a publication.This topic go out of the main purpose of this job.

1.6 The chemical composition prior and after cooking should be of interest

Thank you for your reccomendation. As in the previous point, this topic is outside the scope of the work

Reviewer 2 Report

page 3, lines 103-104: Fish concentrate was the main ingredient studied, therefore the method of obtaining it should be at least briefly described, without the need to read another publication.

page 3, lines 110-111: Table 1 does not correspond to this description, give the exact four recipes in the table; what was used instead of rosemary extract ? why bran was only added to spelt semolina, when bran can also be obtained from durum wheat ? (perhaps the explanation is in earlier studies, but publication should be as much as possible self-explanatory)

page 3, line 115: What is fresh pasta, can frozen pasta be considered as fresh ? Fresh paste is stored and sold at a cold temperature, after being taken out of the packaging it resembles fresh pasta immediately after production. Give some explanation about the types of pasta, as the use of the term "fresh paste" for a frozen product is unclear to me.

page 6, lines 200-204: it is worth giving in chapter 2.1 the basic composition of both flours

Table 3: the two groups (Durum, Spelt) were compared separately (e.g. Durum F - Durum F + antioxidant), the groups should also be compared with each other (e.g. Durum F - Spelt F)

page 7, lines 227-229: What specific ingredients may have had an impact on this?

Author Response

Question

Response

Page

Referee 2

2.1 Fish concentrate was the main ingredient studied, therefore the method of obtaining it should be at least briefly described.

Thank you very much for you reccomendation. A brief description of the procedure has been added to the manuscript.

p. 3

2.2 Table 1 does not correspond to this description, give the exact four recipes in the table; what was used instead of rosemary extract ? why bran was only added to spelt semolina, when bran can also be obtained from durum wheat ?

Thank you for your questions. Table 1 has been completed. There no difference between with antioxidant or without antioxidant. The antioxidant is incorporated in the elaboration of fish concentrate. On the other hand, bran was only added in spelt because the two formulations have different targets. Durum pasta wants to resemble conventional pasta whereas Spelt pasta would be a whole grain option

p. 4

2.3 What is fresh pasta, can frozen pasta be considered as fresh ? Give some explanation about the types of pasta, as the use of the term "fresh paste" for a frozen product is unclear to me.

Thank you very much for your question. According to Real Decreto 2181/1975, fresh pasta is any pasta has not undergone a drying process. This reference has been included in the article.

p. 3

2.4 It is worth giving in chapter 2.1 the basic composition of both flours

Thank you for your advice. We use wheat flour and spelt flour. This is included in that chapter.

p. 3

2.5 The two groups (Durum, Spelt) were compared separately (e.g. Durum F - Durum F + antioxidant), the groups should also be compared with each other (e.g. Durum F - Spelt F)

Thank you very much for your recommendation. We do not compare the two groups because they are different pasta with different composition. Our aim is studied the effect of antioxidant in these types of pasta.

p. 8

2.6 What specific ingredients may have had an impact on this?

Thank you for your question. The main different between formulations is the cereal. This has been included in the manuscript.

p. 8

Reviewer 3 Report

This research investigates the effects of enriched fresh pasta with a sea bass by-product from filleting of sea bass to increase its protein and unsaturated fatty acid contents and to evaluated the stability of polyunsaturated fatty acids during frozen storage for 90 days for consumers. It is an interesting topic for pasta processing industry. Although the use of legume for increasing protein and unsaturated fatty acid contents has been used, the sensory properties of optimal shelf life of enriched fresh pasta made from durum wheat and spelt with by-product from filleting of sea bass are not well evaluated. Sensory profiles of enriched frozen pasta were obtained with addition of a natural rosemary antioxidant. The introduction and materials and methods were well written, nevertheless, there was a few miss-numbering reference and cited references should be corrected. The full journal name of two references (29 & 30) should revise to abbreviation as attached file. The paper is clear and able to read by the focus groups.

The conclusions are well present. Proposed experimental conditions and results could gain high correlated and it is reliable. I suggest some physical methods including cooking loss, texture profile analysis (TPA) and CIE L*a*b* values.

Author Response

Question

Response

Page

Referee 3

3.1 Although the use of legume for increasing protein and unsaturated fatty acid contents has been used, the sensory properties of optimal shelf life of enriched fresh pasta made from durum wheat and spelt with by-product from filleting of sea bass are not well evaluated.

Thank you for your recommendation. We have carried out an analysis of variance to obtain differences between types of pasta and to see where that difference was established. We then perform a QDA (profiling) using square cosines for similarities between pasta types. It is intended to find the influence of the antioxidant for each type of pasta.

p. 5/12/13/17

3.2 There was a few miss-numbering reference and cited references should be corrected. The full journal name of two references (29 & 30) should revise to abbreviation as attached file.

Thank you very much for your notice. The references have been checked. The journal name of these two references have been abbreviated.

p. 18/19/20

3.3 I suggest some physical methods including cooking loss, texture profile analysis (TPA) and CIE L*a*b* values.

Thank you for your suggestion. Cooking loss and CIEL*a*b* values have been included to complete this work. Textura was analysed thoughtout Warner-Bratzler cut test to obtain optimal cooking time.

p.5/6/14/15/17

Round 2

Reviewer 1 Report

I have no more suggestions.